# Dielectric and Energy Storage Properties of Ba_(1−x)_Ca_x_Zr_y_Ti_(1−y)_O_3_ (BCZT): A Review

**DOI:** 10.3390/ma12213641

**Published:** 2019-11-05

**Authors:** Mudassar Maraj, Wenwang Wei, Biaolin Peng, Wenhong Sun

**Affiliations:** 1Research Center for Optoelectronic Materials and Devices, School of Physical Science and Technology, Guangxi University, Nanning 530004, China; mudassar@mail.ustc.edu.cn (M.M.); weiww9189@163.com (W.W.); 2Guangxi Key Laboratory of Processing for Non-ferrous Metal and Featured Materials, Guangxi University, Nanning 530004, China

**Keywords:** dielectric capacitor, lead-free materials, power density, energy storage, multiferroic composites

## Abstract

The Ba_(1−x)_Ca_x_Zr_y_Ti_(1−y)_O_3_ (BCZT), a lead-free ceramic material, has attracted the scientific community since 2009 due to its large piezoelectric coefficient and resulting high dielectric permittivity. This perovskite material is a characteristic dielectric material for the pulsed power capacitors industry currently, which in turn leads to devices for effective storage and supply of electric energy. After this remarkable achievement in the area of lead-free piezoelectric ceramics, the researchers are exploring both the bulk as well as thin films of this perovskite material. It is observed that the thin film of this materials have outstandingly high power densities and high energy densities which is suitable for electrochemical supercapacitor applications. From a functional materials point of view this material has also gained attention in multiferroic composite material as the ferroelectric constituent of these composites and has provided extraordinary electric properties. This article presents a review on the relevant scientific advancements that have been made by using the BCZT materials for electric energy storage applications by optimizing its dielectric properties. The article starts with a BCZT introduction and discussion of the need of this material for high energy density capacitors, followed by different synthesis techniques and the effect on dielectric properties of doping different materials in BCZT. The advantages of thin film BCZT material over bulk counterparts are also discussed and its use as one of the constituents of mutiferroic composites is also presented. Finally, it summarizes the future prospects of this material followed by the conclusions.

## 1. Introduction

The dielectric materials with high energy-storage density, good temperature stability, and low dielectric loss have gained the potential application in pulsed capacitors technology recently. The ceramics industries are trying to develop capacitors with improved energy density, storage efficiency and proper operation sustainability in severe environments [1].

Although electrochemical batteries possess high energy density but their power density is low, on the other side, capacitors usually have high power density while their energy density is small, so they are usually being used to generate a pulsed voltage or current. To enhance the features of current electrochemical energy storage devices the dielectric properties of a capacitor need to be optimized and for that the finding a suitable dielectric material is key for industrial applications [2].

Materials that possess a high electrical resistivity are called dielectrics and can exhibit paraelectric, ferroelectric, and piezoelectric behavior. In the case of paraelectric and ferroelectric behavior, an applied electric field induces a change in polarization, while for piezoelectrics, the effect is due to the application of pressure or temperature, respectively [3]. Ceramics are the prime choice for use in extreme physical conditions such as temperature or pressure if they persist for a long operation time, as compared to metals and plastics. The dielectric properties such as the relative permittivity and dielectric loss factor of ceramic material are affected significantly as frequency changes, therefore their ranges have to be determined before the design of any system containing their utilization. Thus, research on ceramic technology is currently flourishing in order to obtain advanced materials with required properties and rapid demand [1].

In this review we will specifically focus on the lead-free Ba_(1−x)_Ca_x_Zr_y_Ti_(1−y)_O_3_ (BCZT) material, its dielectric and energy storage properties and future applications. The BCZT is a lead-free piezoelectric ceramic with a perovskite structure and has attracted more attention than its counterpart lead zirconate titanate (PZT) due to its superior piezoelectric properties, low *P_r_* value and ease of controlling the properties by simply changing the concentration of Ca and Zr contents in Ba_(1−x)_Ca_x_Zr_y_Ti_(1−y)_O_3_ composition. By focusing the dielectric properties mainly in our review, we will provide the conditions under which these parameters vary to meet the certain requirements. Firstly, we will discuss the dielectric properties of a capacitor followed by a discussion on different ceramic materials possessing extraordinary dielectric properties and after that the different synthesis technique outputs will be discussed, accompanied by a description of dielectric properties of both pure and doped BCZT. BCZT thin films dielectric properties are also discussed. The BCZT usage as ferroelectric constituent in multiferroic composite materials is also described and at the end future perspectives of BCZT research are discussed followed by the concluding remarks.

## 2. An Overview of Dielectric Properties

### Dielectric Properties and Energy Storage

Dielectrics can be broadly grouped into being either linear or non-linear, according to the relationship between the applied electric field and the polarization. For linear dielectrics the energy density (*U*) is generally expressed as *U* = *1*/*2 ε*_0_*ε_r_E*^2^. For the non-linear dielectrics, the polarization can be determined from the polarization-electric field (P-E) hysteresis measurements by the following equations
(1)$$P=\varepsilon_{0}\left(\varepsilon_{r}-1\right)E=\chi_{e}\varepsilon_{0}E$$
where *P* is polarization and χ is dielectric susceptibility. Energy density, *U*, is a measure of the energy stored per unit volume. For dielectrics, this can be obtained by the following relationship:(2)$$U=\int_{0}^{Emax}PdE.$$

Using Equation (2), the *U* values of the dielectrics can be obtained through the numerical integration of the area between the polarization and curves for the P-E loops [2]. It can be concluded from the above equations that the larger energy densities can be obtained by greater relative permittivity, highest polarization, and breakdown strength (BDS), while low dielectric losses, reduced remnant polarizations, and smaller value of loss tangent (*tanδ*) will ensure greater energy storage efficiencies of dielectric materials. In this review we will describe the work by different authors to optimize these properties under different conditions which will lead to future applications of BCZT in industry.

## 3. Dielectric Materials for High Energy-Storage Application

To design a good dielectric material having a high energy-storage density and higher efficiency for required application, thehigh breakdown strength (BDS) [4], large saturated polarization, and small remnant polarization [5] should be satisfied simultaneously [2]. As there are four main categories of dielectric materials, namely linear dielectric, ferroelectric, relaxor ferroelectric (RFE), and antiferroelectric, comparatively, the only two of them, the relaxor ferroelectrics and antiferroelectrics, are more suitable candidates for high energy storage applications because of their higher saturated polarization and lower remnant polarization. Each of them are described by the P-E loop given in Figure 1 [2]. Furthermore, the RFE dielectrics are further categorized as lead-based and lead-free RFE dielectrics. The following sub-sections describe dielectric properties of both of them and we will focus mainly our lead-based RFE ceramic, the BCZT.

### 3.1. Lead-Based RFE Ceramics

The RFE ceramics are more attractive for energy storage applications due to their outstanding dielectric and ferroelectric properties. These materials exhibit higher relative permittivities and saturation polarizations but slimmer hysteresis loops, which are one of the pre-requisites for realizing high energy densities and energy storage efficiencies. RFE are categorized as two main types, lead-based and lead-free relaxors [2].

There are a few lead-based relaxor ceramics which were examined for energy storage applications such as PbZrO_3_-(SrTiO_3_) [6], Pb(Mn_1/3_Nb_2/3_)O_3_-PbTiO_3_ [7,8,9], Eu-doped PbZrO_3_ [10], Pb(Ni_1/3_Nb_2/3_)0.5Zr_0.15_Ti_0.35_O_3_ (PNNZT) [11], Pb_0.8_Ba_0.2_ZrO_3_ [12], La-doped Pb(Zr,Ti)O_3_ [13,14,15], Pb[(Ni_1/3_ Nb_2/3_)0.6Ti_0.4_O_3_ (PNNT) [16], and Pb_0.8_Ba_0.2_ZrO_3_ (PBZ) [17,18], Pb_0.99_Nb_0.02_(Zr_0.65_Sn_0.3_Ti_0.05_)_0.98_O_3_ [19], and Pb_0.78_Ba_0.2_La_0.02_ZrO_3_ [20]. Among all the lead-based ceramics the most widely used piezoceramic material is Lead zirconate-titanate (Pb(Zr,Ti)O_3_ (PZT)) but due to the toxic environmental issues, the European Union introduced a directive for the use of lead-based elements in commercial products and these regulations have attracted researchers to explore lead-free piezoelectrics [21,22,23].

### 3.2. Barium Titanate (BT)-Based Lead-Free Ceramic System

The BT-based ceramic system with an ABO_3_ perovskite structure (A is calcium, strontium, barium, etc., and B is tantalum, titanium, etc.) [24], has advantages of low dielectric loss and stable electrical properties for operations of long durations, but their use is limited due to low *T_c_* and *d_33_* even at room temperature [25]. BaTiO_3_ is the first reported lead-free piezoelectric ceramic and an important ferroelectric material used in many processes and applications in electronics, due to its typical perovskite structure and specific characteristics, such as a high dielectric, ferroelectric, and piezoelectric properties. Figure 2 describes the unit cell and crystal structure of BaTiO_3_.

BaTiO_3_-based RFE ceramics are considered as one of the most favorable materials for energy storage applications. The mostly explored perovskite-based lead-free piezo-ceramics are BaTiO_3_ (BT), Bi_0.5_Na_0.5_TiO_3_ (BNT), and K_0.5_Na_0.5_NbO_3_ (KNN) [26,27,28,29,30,31,32,33,34,35,36,37,38,39,40,41,42]. Ogihara et al. [43] described the energy storage properties of 0.7BaTiO_3_-0.3BiScO_3_ (BTBS) ceramics. Since then, many researchers have investigated various BT-based RFE materials such as BaTiO_3_-Bi(Mg_2/3_Nb_1/3_)O_3_, BaTiO_3_-Bi(Mg_1/2_Nb_1/2_)O_3_, BaTiO_3_-Bi(Zn_2/3_Nb_1/3_)O_3_, BaTiO_3_-Bi_0.5_Na_0.5_TiO_3_-Na_0.73_Bi_0.09_NbO_3_ [18], and Ba(Zr_0.2_Sn_0.2_Ti_0.2_Hf_0.2_Nb_0.2_)O_3_ [44]. 

### 3.3. Modified BaTiO_3_ (From BaTiO_3_ to BCZT)

BT-based ceramics, however, exhibit comparatively low Curie temperature *T_c_* (~120 °C) thereby limiting the working temperature range and their use in piezoelectric applications [26]. So researchers are trying to improve their piezoelectric properties. Using a sol–gel method, Praveen et al. [45] found that the better control of phase composition of BZT–BCT give rise to lower processing temperatures and finer particle size. The resultant material had a high *P**_r_* of 12.2 mC cm^−^^2^, a high *d_33_* of 637 pCN^−^^1^, and a large *k**_p_* of 0.596. The formation of morphotropic phase boundary (MPB) improved piezoelectric properties in the doped ceramics. Liu and Ren [46] reported a new lead-free system, Ba(Zr_0:2_Ti_0:8_)O_3_x_(Ba_0.7_Ca_0.3_)TiO_3_ (BZT-xBCT), characterized by a phase boundary between rhombohedral (BZT side) and tetragonal (BCT side) phases [47,48,49,50,51,52]. Since then, attention has been directed towards Ba(Ti_0.8_Zr_0.2_)O_3_(Ba_0.7_Ca_0.3_)TiO_3_ (BCZT) because it has a large piezoelectric coefficient (*d_33_* > 500 pC/N but low *T_c_*) at high sintering temperature ≈ 1500 °C [46]. Furthermore, BCZT is an environmentally friendly material because it is a lead-free piezoelectric ceramic. Nevertheless, the pure perovskite phase for BCZT can only be achieved at very high calcination and sintering temperatures, which is extremely high for some of the practical applications. They also observed the presence of a Cubic-Rhombohedral-Tetragonal (C-R-T) triple point at about x = 0.32 and at 57 °C, which had been widely reported in Pb-based ferroelectric materials [53]. They reported the highest spontaneous polarization, *P_s_* ≈ 20 µC-cm^−2^, the highest remnant polarization *P_r_* ≈ 15 µC cm^−2^, the lowest coercive field *E_c_* ≈ 0.168 V mm^−1^ and the highest relative permittivity ε ≈ 3060 for x = 0.5 (denoted 50BCT). The properties of these materials were said to be comparable to those reported for Pb-based systems. The outstanding properties of the x = 0.5 composition reported by Liu et al. [46] attracted further research to explore the BCZT. Figure 3 shows supercell construction and arrangements of different atoms in a BCZT crystal [54]. 

## 4. Dielectric and Energy Storage Properties of Pure BCZT

Since the work of Liu et al., efforts are still ongoing for the enhancement of the relaxor behavior of BCZT materials, enhancing the energy storage density, storage efficiency, improving the dielectric constant, and reducing the dielectric loss. Remarkably, Z. Hanani et al. worked on the dielectric constant and dielectric loss for different frequencies as shown in the Figure 4 and also presented a remnant polarization *P_r_* of ~2.03 μC/${\mathrm{cm}}^{2}$ and coercive field $E_{c}$ of ~2.17 kV/cm at 0 °C. 

They found that as the temperature increased, the P–E loops had become slimmer and there was a continuous decrease of *P_r_*, *E_c_*, and *P_max_* [55]. The W_rec_ (green area of Figure 5a) increased gradually with temperature and reached a maximum of ~14 mJ/cm^3^ at 90 °C and then decreased.

Jiagang Wu et al. reported that the increase of sintering temperature causes the relative density and average grain size of BCZT to gradually increase, and electrical properties are also enhanced. They showed that the $\varepsilon_{r}$ value of BCTZ ceramics increased and the *tanδ* decreased with an increase of sintering temperature [56]. The Ca and Zr contents also matter a lot regarding dielectric properties as described by Ye Tian et al. They found that that the $\varepsilon_{r}$ increased with the increase in Ca content and the highest value of $\varepsilon_{r}$ (= 4821) was found at x = 0.15. The $\varepsilon_{r}$ increases significantly at y = 0.10 and reaches maximum at y = 0.15, as shown in Figure 6 [57]. It can be seen from Figure 6a that by increasing the Ca contents, the material at morphotropic phase boundary (MBP) region, entered to diffuse ferroelectric phase transition region (DPTR) while this region was not achieved by changing Zr concentration.

The dielectric properties of Ba_0.95_Ca_0.05_Zr_0.3_Ti_0.7_O_3_ were determined as a function of temperature, frequency and electric field by Di Zhan et al. They achieved an energy storage density of 0.59 J/${\mathrm{cm}}^{3}$ and storage efficiency of 72.8% [58]. The grain size also affected the dielectric properties of BCZT significantly as parallel to sintering temperature as described by X. G. Tang et al. They investigated that the ε_m_ decreased as the grain size decreased. They predicted that by varying the ratio of Zr/Ti, the dielectric constant could be adjusted to appropriate values for tunable capacitor applications [59]. The improved dielectric properties were obtained by Z. Hanani et al. with the control of grain size. The dielectric constants were found to be in the range of 5370 to 9646 and the *tanδ* was enhanced by 70%. Their results showed that there existed an undisputable link between mean grain size with dielectric properties [55].

## 5. Synthesis Techniques Effects on Dielectric Properties of BCZT

Synthesis routs and characterization tools are crucial aspects to a functional material. The preparation techniques significantly alter the dielectric properties of a materials so in this section we will review different synthesis methods for BCZT ceramic material synthesized by different methods, and in each method changing the sintering temperature leads toward different values of the dielectric properties. Firstly, we will discuss the most widely used method, sol–gel method preparation of BCZT. Sol–gel method preparation of BCZT material at different temperatures was discussed by W. Li et al. and they employed this method on two different substrates [60]. They found the high tunability of 64% for the LNO/Pt/Ti/SiO_2_/Si substrate with the sol–gel method and concluded that the BCZT is a promising candidate for MEMs applications. Sadhu et al. also used sol–gel method and focused the energy storage density and obtained energy density of 8.5 j/cm^3^ for BCZT-Poly(vinylidiene fluoride-hexafluoropropylene) BCZT/PVDF-HFP composite films with 10 μm thickness [61]. They evaluated the dielectric properties with different filler concentrations and found the lower dielectric loss of 0.036 and BDS of 265 KV mm^−1^. We also used the sol–gel method and demonstrated a high energy storage density (W∼15.5 J/cm^3^) and ultra-high efficiency (η ∼93.7%) of La doped BCZT thin films [62]. BCZT ceramic sintered at 1400 °C had a uniform microstructure with a density of 5.56 g/cm^3^ and possessed excellent dielectric properties of ε_m_ = 8808, 2*P_r_* = 24.48μC/cm^2^, *d_33_* = 485 pC/N as evaluated by X. Ji et al. [63]. Zhong Ming Wang et al. indicated that BCZT ceramics obtained by sol–gel method had outstanding electrical and energy storage properties (*d_33_* = 558 pC/N, kp = 0.54, *P_r_* = 12.15 µC/cm^2^, ε_m_ = 16,480, *ε_r_* = 3375, *tanδ* = 0.021, and W = 0.52 J/cm^3^ [64]. X. Yan et al. synthesized BCZT by the sol–gel technique sintered under different sintering routes. It was found that the sintering rate of the nanocrystalline powders was significantly improved in the case of the conventional sintering conditions. The sintering temperature was reduced from 1540 to 1280 °C [65]. 

The conventional solid state method had attracted lot of attention in fabrication of BCZT material. Proper sintering temperature will lead to crystallinity and excellent elemental composition, moreover good crystal lattice and plane spacing. B. Luo et al. synthesized BCZT with the conventional solid state method and concluded that for the BCZT thin films fabricated by this method, the *ε_r_* increased with the increase of BCZT contents, whereas the *tanδ* remained constant for a given frequency range [54]. X. Chao et al. also used this method to get, ε_m_ = 13,298, *ε_r_* = 3375, *tanδ* = 0.021, *P_r_* = 8.17 µC/cm^2^, *E_c_* = 3.45 kV/cm. The *ε_r_* = 3762 and *tanδ* = 1.05% were obtained by Y. Cui et al. using this method [66]. A. Li et al. co-doped La/Y with BCZT by this method and evaluated a large permittivity (21,736) by using this method [67]. Q. Li et al. obtained excellent electrical behavior with excellent piezoelectric coefficient (*d_33_* = 554 pCN^−1^), dielectric constant (ε = 4958), and dielectric loss (*tanδ* = 1.8%) when sintered at 1450 °C [68]. Y. Zhang et al. used solid state method for Mn-doped BCZT and obtained a dielectric constant of (*ε_r_* = 1580) and a low dielectric loss of (*tanδ* = 2.94%) [52]. V_2_O_5_ was doped with BCZT by Y. Yang et al. and solid state method yielded *d*_33_ = 466 pC/N, *P_r_* = 7.735 μC/cm^2^, *ε*_r_ = 3104, and *tanδ* = 0.03, implying favorable applications for lead-free piezoelectric ceramics [50]. Some other methods like surfactant-assisted solvothermal processing [55,56,57,58,59,60,61,62,63,64,65,66,67,68,69] oxide-mixed method [56], citrate method [51,52,53,54,55,56,57,58], ceramic route [51], mixed oxide (and carbonates) method [70,71,72], spark plasma sintering technique [73], and Pechini polymeric method [74] were also employed.

Concludingly, by changing simply the sintering temperature, the g different values of dielectric constant, dielectric losses, and energy density can be obtained in a single synthesis method. Increasing the sintering temperature causes enhanced grain size and dielectric properties. Therefore, by varying sintering temperature and fabrication method different results of BCZT properties can be extracted using sol–gel, mixed-oxide, solid state, molten salt and solvothermal methods etc. but among these the sol–gel method offers the opportunity to mix metal ions at atomic level, lower reaction temperature, ultra-homogenous mixing power and provides better energy density due smaller grain size and uniform structure of particles.

## 6. Dielectric Properties of Doped BCZT

For certain applications, the ferroelectrics are rarely used in a chemically pure form and doping is usually employed to achieve the desired results. Different doping usually brings about different consequences, such as acceptor doping to obtain low dielectric losses and donor doping to achieve high piezoelectric coefficients [75]. Therefore, in this section we will see how different doping results in different values of dielectric properties of BCZT. S. D. Chavan et al. investigated the dielectric constant of composites La_0.67_Sr_0.33_MnO_3_-Ba_0.95_Ca_0.05_Ti_0.90_Zr_0.10_O_3_ (LSMO-BCZT) for the frequency between 20 Hz and 1 MHz as a function of applied magnetic field and observed that $\varepsilon_{max}$ varies from 767.75 to 453.33 for the frequency from 100 Hz to 1 MHz so BCZT showed relaxor behavior as shown in Figure 7 [51].

D. Radoszewska et al. modified BCZT ceramics by an admixture of strontium and found that the rise of the dopant concentration led to the increase of the dielectric permittivity peak value, with a simultaneous shift of the corresponding temperature to lower values. However, obtained ceramic materials showed very small temperature hysteresis (about 4 K). The increase of the strontium concentration in the basic ceramics also influenced the shape of temperature dependence of the loss tangent (*tanδ*) [51]. 

Xiaolian Chao et al. synthesized (1−x)(Ba_0.85_Ca_0.15_)(Zr_0.1_Ti_0.9_)O_3_–xBiAlO_3_ ((1−x)BCZT–xBAO) piezoelectric ceramics and investigated the effects of BAO and found the decrease in the sintering temperature from 1450 to 1300 °C. They obtained the dielectric properties, $k_{p}$ = 0.54, $P_{r}$ = 8.17 lC/cm^2^, $E_{c}$ = 3.45 kV/cm, $\varepsilon_{m}$ = 13,298, $\varepsilon_{r}$ = 3375, *tanδ* = 0.021 for x = 0.8 mol% [76]. Figure 8 shows their results for varying the dopant concentration. Yerang Cui et al. fabricated lead-free (Ba_0.7_Ca_0.3_)TiO_3_-Ba(Zr_0.2_Ti_0.8_)O_3_-xwt%CuO(BCZT-xCu) piezoelectric ceramics and studied the effects of CuO addition, obtaining a high dielectric constant $\varepsilon_{r}$ = 3762 and a *tanδ* = 1.05. The dielectric properties of BCZT-xCu ceramics were also measured as a function of temperature. A low *tanδ* and high $\varepsilon_{r}$ indicate that the BCZT-xCu ceramics could also be a promising lead-free candidate for practical applications [66]. 

E. El. Shafee et al. described the properties of Ba_0.95_Ca_0.05_Ti_0.8_Zr_0.2_O_3_–poly(vinylidene fluoride-trifluoroethylene) BCTZ-P(VDFTrFE) composite and showed that the dielectric properties changed greatly with frequency. The values $\varepsilon_{r}$ and *tanδ* of these composites reduce more quickly with frequency at low frequencies ~1 kHz [77]. 

P. Jaimeewong et al. investigated the effect cobalt oxide (CoO) and iron oxide (Fe_2_O_3_) as dopants on dielectric properties of BCZT and found that the BCZT-0.01Co ceramics presented the finest properties, that is, $d_{33}$ = 276 pC/N and $\varepsilon_{r}=$ 4567. The Co doped BCZT ceramics could be employed to prepare lead-free materials with striking properties as shown in the Figure 9 [71]. 

Anqi Li et al. investigated the dielectric properties of Lanthanum(III) oxide $\left(La_{2}O_{3}\right)$ and Yttrium oxide $\left(Y_{2}O_{3}\right)$ co-doped (Ba_0.9_Ca_0.1_)(Zr_0.2_Ti_0.8_)O_3_ ceramics and found a large permittivity (21,736) and low loss tangent of 0.063 [67]. Q. Li et al. discussed the properties of *Sm*_2_*O*_3_ doped Ba_0.85_Ca_0.15_Ti_0.90_Zr_0.10_O_3_ (BCTZ–x*Sm*) ceramics, and observed piezoelectric and dielectric properties. Their results revealed that the addition of *Sm*_2_*O*_3_ improved the piezoelectric property of BCZT ceramics significantly while affecting the Curie temperature ($T_{c}$) negatively [68]. Jiafeng Ma et al. investigated the microstructures, phase structure, and electrical properties of the *Sb*_2_*O*_3_-modified BCZT lead-free piezoelectric ceramics. The grain size was strongly affected by *Sb*_2_*O*_3_. The ceramics with x = 0.1% exhibited the best dielectric properties of $\varepsilon_{r}$ = 3895, *tanδ* = 1.3%, and $P_{r}$ = 12.6 µC/${\mathrm{cm}}^{2}$ as shown in the Figure 10 [78]. Wu et al. fabricated BCTZ + x mol.% ZnO ceramics with x = 0, 0.02, 0.04, 0.06, 0.08, and 0.10 and found that with the increase of x to 0.1%, the ZnO-modified BCTZ, the $\varepsilon_{r}$ value first increases and then increases slightly at x ≥ 0.1% [79]. 

P. Parjansri et al. investigated the dielectric properties of $Nb^{5+}$ (0.0–1.0 mol%) doped with Ba_0.90_ Ca_0.10_Zr_0.10_-Ti_0.90_O_3_. The highest value ~4636 of dielectric constant was found at 1.0 mol% Nb. The dielectric loss was less than 0.03 for all samples at room temperature [80]. Yixuan Yang et al. studied the effects of $\;V_{2}O_{5}$ doping on the phase composition, microstructure, and electrical properties of the lead-free BCZT. The dielectric properties of BCZTxV ceramics obtained as, $P_{r}$ = 7.735 μC/${\mathrm{cm}}^{2}$, $\varepsilon_{r}$ = 3104, and *tanδ* = 0.03 at x = 0.2 mol% [50]. Ferroelectrics (Ba_0.92__−x_Ca_0.08_*Nd*x)(Ti_0.82_Zr_0.18_)O_3_ (0 ≤ x ≤ 0.02) were prepared by P. Yong et al. and they found that with increasing *Nd^3^**^+^* concentration, the dielectric constant decreased from 17,184 (x = 0) to 6095 (x = 0.02), while the dielectric loss decreased from 0.0245 (x = 0) to 0.0119 (x = 0.02) simultaneously [81]. Yong Zhang et al. fabricated pure and Mn-modified Ba_0.95_Ca_0.05_Zr_0.1_Ti_0.9_O_3_ ceramics and studied the influence of Mn and sintering temperature on dielectric properties of BCZT. As compared to undoped BCZT, 1.2 mol% Mn-modified ceramics had better dielectric properties and a low dielectric loss. The dielectric loss of pure ceramics was found to be higher than that of Mn-doped BCZT ceramics [52]. The abovementioned details are summarized in Table 1.

## 7. BCZT Thin Films for Enhancement of Dielectric Properties

Due to the environmentally friendly nature of thin film processing, the technology is transferring to micro-scale devices. Therefore, the thin films of the lead-based ferroelectrics are gaining attention of researchers these days [82,83]. Recently, many efforts have been given to the practical applications of thin films for electronic devices. The thin films exhibit versatile physical properties with respect to the bulk counterpart [84]. Different physical and chemical methods such as PLD, RF-magnetron, sol–gel, CSD etc., have been used for lead-free ferro/pizoelectric thin films depositions and in each case the substrate matters a lot [85]. By changing the preferential orientation and substrate layers we can achieve different dielectric properties. The following paragraphs will shed light on some efforts to improve the dielectric properties of BCZT thin films.

Shihua Yang et al. prepared polycrystalline Ba_0.85_Ca_0.15_Ti_0.9_Zr_0.1_O_3_ (BCZT) thin films with random orientation and (100) preferential on the LaNiO_3_ electrodes. Their results showed that the dielectric constant and tunability of the random orientation films are smaller than those of the (100) preferential film. The BCZT thin films led to a larger dielectric constant and higher dielectric tenability at the frequency of 100 kHz in the (100) preferential film [86]. L.N. Gao et al. deposited Ba_1−x_CaxZr_0.05_Ti_0.95_O_3_ (x = 0, 0.05, 0.10) (BCZT) thin on Pt/Ti/SiO_2_/Si substrates by sol–gel processing and reported that the average grain size reduced with an increase in the Zr content, which also decreased the dielectric constant and maintained a leakage current low and stable. The frequency dependent dielectric constant of their films is shown in Figure 11 [87]. 

Wei Li et al. measured the dielectric constants as a function of temperature for BCZT thin films deposited on Pt/Ti/SiO_2_/Si and LNO/Pt/Ti/SiO_2_/Si substrates. The BCZT films deposited on Pt/ Ti/SiO_2_/Si had dielectric constant 550 and that of LNO/Pt/Ti/SiO_2_/Si substrates had 620, respectively (Figure 12). The films deposited on LNO/Pt/Ti/SiO_2_/Si substrates showed a high tunability of 64%. Their results indicated that the BCZT thin films are good candidates for the lead-free MEMs application [60]. 

Thin-film capacitor stacks were fabricated for BCZT thin films with sputtered Pt and/or Ni electrodes by G. J. Reynolds et al. and they obtained the maximum energy density of about ~4.7 × 10^−2^ W-h/liter for these capacitors which corresponded to a specific energy of ~8 × 10^−3^ W-h/kg [88]. José P. B et al. investigated the effect of thin dielectric layer of HfO_2_:Al_2_O_3_ (HAO) on the energy storage properties in the Pt/0.5Ba(Zr_0.2_Ti_0.8_)O_3_–0.5(Ba_0.7_Ca_0.3_)TiO_3_ structure. Energy density of Pt/BCZT/HAO/ Au capacitors was found to be 99.8 J cm^−3^ and efficiency was 71.0% [89]. K. Mimura et al. fabricated BZT and BCZT thin films on Pt/TiOx/SiO_2_/Si substrates. They found that, Ca-doped BZT thin films had a larger dielectric constant than a BZT film without Ca (Figure 13) [90]. 

L.L. Jiang et al. grew (BCZT) thin films on Pt/Ti/SiO_2_/Si substrates with and without a CaRuO_3_ (CRO) buffer layer. The dielectric constant and tunability were 725 and 47.0%, 877 and 50.4%, respectively, for substrates without and with the CRO buffer layer. A higher dielectric constant and tunability were observed for Pt/BCZT/CRO capacitors [91]. C. J. M. Daumont reported on libraries of Ba_0.97_Ca_0.03_Ti_1_x_Zr_x_O_3_ thin films deposited on IrO_2_/SiO_2_/Si substrates and found maximum tunabilty at intermediate compositions of 60% for an electric field of about 400 kV cm^−1^ [92]. Min Shi et al. prepared the lead-free thin films of BCZT on the Pt/Ti/SiO_2_/Si for different annealing processes. The annealing process, where the thin film is composed of a single phase with no trace of secondary phases with preferred orientation of (100), was found to be the best for preparing thin films of BCZT with good dielectric properties [93]. Valentin Ion et al. showed nontrivial dependence on film thickness for epitaxial thin films of Ba(Ti_0.8_Zr_0.2_)O_3_-x(Ba_0.7_Ca_0.3_)TiO_3_ (x = 0.45) ceramics and showed that the dielectric constant reached the highest value of around about 3400. The dielectric loss was small (nearly 1%) although it started to increase slightly above 100 kHz [94]. Mg-doped BCZT thin films were fabricated by Kalkur et al. The capacitors which were post annealed at 700 °C showed enhanced dielectric properties. The Pt/BCZT/Pt/MgO capacitors exhibited high tenability of 55% at an applied field of 55 kV/cm. Therefore, for the fabrication of tunable devices, the BCZT could be a promising high dielectric constant candidate material [95]. V S Puli et al. investigated BZT–BCT films at the vicinity of the morphotropic phase boundary (MPB) of the [(1 − *x*) BZT–*x*BCT] (*x* = 0.50) composition [96]. Their BZT–BCT films exhibited a high saturation polarization of 148 *μ*C cm^−2^ and a high energy-storage density of 39.11 J cm^−3^ at 2.08 MV cm^−1^ [97].

Depending on the materials involved, the thin films are categorized as organic, inorganic, and composite organic-inorganic thin films. The advantage of inorganic thin film over organic counterparts is that they show durability, resistance to harsh environmental conditions, and stability against degradation. On the other hand, organic materials are usually flexible, transparent, and are likely to degrade when exposed to heat, moisture, oxygen, etc. Organic-inorganic composite films have attracted great attention because of the potential of combining properties of organic and inorganic components within a single molecular composite. These composite films are considered as potential candidates for different future applications such as in transportation, micro-electronics, health, energy, and diagnosis. The properties of these films can be tuned by modification of the composition of the elements to produce smart materials. Therefore, this composite nature is also being utilized for improving the dielectric properties of BCZT. H. W. Lu et al. fabricated the PI@BCZT/polyvinylidene fluoride (PVDF) flexible composite films and their results showed that the 50 vol% composites possessed an enhanced dielectric permittivity (*ε_r_* = 130) at 100 Hz while the dielectric loss was reduced from 1.8 to 0.2 while breakdown strength had increased from 20 to 96 kV mm^−1^ [98].

BCZT/PVDF composite flexible films with excellent dielectric properties were fabricated with dopamine@BCZT powders as fillers by Luo et al. and they observed increased *ε_r_* with the increase of dopamine@BCZT content (Figure 14). The energy storage density was about 2.0 J cm^−3^ for the volume fraction 0.61 [54]. 

Sai Pavan et al. studied the 0–3 type BCZT-Poly(vinylidiene fluoride-hexafluoropropylene) (BCZT/PVDF-HFP) polymer nanocomposites for dielectric properties and energy storage densities. They observed maximum energy storage density of 8.5 J cm^−3^ for filler concentration of 10 vol% for films of 10 μm thickness. Figure 15 shows their dielectric properties with different filler concentrations [61]. The NiFe_2_O_4_/Ba_0.85_Ca_0.15_Ti_0.9_Zr_0.1_O_3_ (NFO/BCZT) composite thin films were grown on (100)-SrRuO_3_/SrTiO_3_ substrate by Dai et al. The *P_s_* and *P_r_* of NFO/BCZT film were about 35.79 and 12.43 μC cm^−2^ respectively. The saturated and remnant polarizations of BCZT (*P_s_* = 57.88, *P_r_* = 19.45, μC cm^−2^) were higher than that of NFO/BCZT. It had been reported that doping with Ca^2+^ at A-site and substitution of B-site with Zr^4+^ could give rise to a lower dielectric loss and larger *d_33_* [99,100,101,102].

Recently we achieved large energy storage density (W∼15.5 J/cm^3^), ultra-high efficiency (η∼93.7%), and high thermal stability in the La-doped BCZT (Figure 16). The thin films deposited on Pt bottom electrodes exhibited a larger ε value and a smaller *tanδ* value than the corresponding films deposited on LaNiO_3_/Pt composite bottom. The largest W (15.5 J/cm^3^) was obtained for x = 0.0075 in the La-doped thin film deposited on the LaNiO_3_/Pt electrode (Figure 17) [62]. The information about thin films dielectric properties are summarized in Table 2 below.

### 7.1. BCZT for Multiferroic Composites

The multiferroic materials are the materials with coexistence of two or more of ferroelectricity, ferromagnetism and ferroelasticity simultaneously in a single phase as shown in Figure 18 and have become a more interesting field of research for the scientific community recently for their unusual physical properties [103]. 

As there is a coupling between ferroelectric and ferromagnetic properties in these materials, they are highly important for different future applications, such as stretchable antenna, magnetic memory elements, microelectronic devices, spintronics, information storage, signal processing, transducers and other compact size components. For further readings about multiferroic composite materials in detail, we refer the readers to the different reviews [104,105,106].

The dielectric properties of some lead-based NiPZT, NZFO-PZT [107], PMN-PT [108], and PZT-CFO [109] as well as lead-free CoFe_2_O_4_ (CFO)\BaTiO_3_ (BTO ) [110,111], (1−x)BNT-BT-x NCFZ) [112], (1−x)BaSrTiO_3_ + xNiCoFe_2_O_4_Bi_4_ [113], BiNdTiFeCoO_15_ [114], BaTiO_3_-(Ni_0.8_Zn_0.2_)Fe_2_O_4_ [115], ZnFe_2_O_4_-N_0.5_Bi_0.5_TiO_3_ [116], NiCoMnFeO_4_\NaBiTiO_3_ (NMF\NBT) [117], and BNKT-BNiJ [118] multiferroic composites are also studied. The BCZT is also being employed in multiferroic composites as a piezoelectric constituent and has been explored with different concentrations of Ca and Zr elements. N.S. Negi et al. calculated dielectric properties of (BCTZ-CFO) multiferroic composites and found 10 wt. % of CFO provided maximum remnant polarization of *P_r_* ~ 1.4 μC/cm^2^. The mixture of CFO and BCTZ decreased the dielectric constant and increased the loss tangent values. The higher values of dielectric constant were obtained at lower frequencies while the loss *tanδ* was high at low frequency [119]. N. Shara Sowmya1 et al. investigated ferroelectric and ferromagnetic properties 50BCZT-50NFO multiferroic composite and found that the remnant and saturation polarizations of 8.34 and 17.4 μC/cm^2^, respectively. They concluded that the composites comprised of the ferrite and ferroelectric phases, affected electric and magnetic hysteresis loop trends significantly. The M-H and P-E hysteresis loops of these composite exhibited improved coercivity and *P_r_* [120]. S.B. Li. et al. investigated the (1−x)Ba_0.85_Ca_0.15_Zr_0.10_Ti_0.90_O_3_-xLa_0.67_Ca_0.33_MnO_3_ ((1−x)BCZT-xLCMO), and (BCZT/LCMO/BCZT) laminated composites and observed the improved dielectric properties for these laminated composites as compared to that of pure BCZT. The dielectric properties were measured for both different temperatures (25–200 °C) and frequencies (20 Hz–1 MHz). A continuous decrease in *ε_r_* was observed in both composites with increasing frequency whereas the loss tangent initially decreased and then increased. By increasing the LCMO content, the *ε_r_* and *tanδ* increased. The highest values of ε_m_ was found to be 20,280 for x = 0.5 which indicated that the dielectric constant strongly depends on the thickness ratio between ferromagnetic LCMO and the ferroelectric BCZT (Figure 19) [121,122]. 

Paul Praveen et al. explored BCZT-CoFe_2_O_4_ multiferroic particulate ceramic composite and found a high dielectric constant of 2160 and tangent loss factor of 0.6. Tangent loss (*tanδ*) also showed a similar behavior and a maximum loss of 0.6 has been obtained for the composite. The hysteresis loop was slim and indicated the low resistive nature of the composite [123]. 

Jian-Min Yan et al. used (BZT-BCT) piezoelectric ceramic substrates for the deposition of ferromagnetic (LSMO) polycrystalline films for lead-free multiferroic structures and found that the dielectric constant was maximum for smaller frequencies and decreased with the increase in frequency [124]. Md Abdullah-Al Mamun et al. investigated lead-free BZT–BCT/LSMO for multiferroic applications. The *ε_r_* varied from 5100 to 4900 for the frequency range of 1–50 kHz. The dielectric loss was observed as low as (0.02) so ensures the extraordinary quality for ferroelectric device fabrication. The dielectric constant decreased with the increasing frequency whereas the *tanδ* increased. The magnetocapacitance was calculated to be 2.8 nF corresponding to high dielectric constant of ~5100 at 1 kHz. These results affirmed that these materials could be employed in actuators and next-generation sensors by considering their good magnetoelectric and electromechanical coupling properties of the lead-free FE/FM heterostructures [125]. M. Naveed-Ul-Haq et al. studied the dielectric properties of BCZT-CFO(CoFe_2_O_4_) multiferroic composite and showed that the temperature dependence of the real part of electric permittivity *ε′(T)* for BCZT85-CFO15 composite exhibited a peak value at around 370 K. Additionally, the permittivity continuously increased with increase in temperature at lower frequencies. Their composite sample had strong frequency dispersion. They also found *ε′(f)*, frequency dependence of permittivity and found that the ferrite phase partly affected the dielectric response of the composite. They considered BCZT85-CFO15 composite as the best choice for future lead-free materials with higher magnetoelectric properties [126]. Sagar Mane et al. prepared multiferroic nanocomposites of x[CoNiFe_2_O_4_]-(1_x)[0.5(BCT-BZT], x = 0.1, 0.2, 0.3, 0.4, and studied the *ε_r_* and *tanδ* and loss factor as a function of frequency and temperature of the composites. The dielectric constant of the ferroelectric phase was found to be larger than that of ferrite phase at frequency above 1 KHz. The enhanced magnetocapacitance was measured with increase in the magnetic field. The dielectric constant decreased with the increase in frequency, while at higher frequencies it remained constant. On the other hand, as the frequency increased, *tanδ* decreased. The enhancement of ferrite concentration causes an increase in the dielectric loss. The maximum electric permittivity was found for 10CNFO-90[50BCT-BZT] concentration of around 2264 [127]. Magnetodielectric properties of LSMO and BCT-BZT composites were also reported by S S Mane et al. The temperature range of 30–150 °C and frequencies of 1, 10, 100 kHz, and 1 MHz were chosen for dielectric properties measurement. It was observed that the *T_m_* shifts slightly toward higher temperature side as frequency increased from 100 Hz to 1 MHz. The *tanδ* was found to be maximum at *T_m_* for the composition 0.50BCT–0.50BZT. With the increase in frequency, the dielectric loss first decreased then nearly reached a constant value [128]. Pakpoom Jarupoom et al. investigated the effects of LSMO additive on the dielectric properties of BZT ceramics. The addition of LSMO caused change in the dielectric permittivity curves. The *tanδ* increased with increase in LSMO content, and it was suggested that introducing LSMO has a strong effect on the dielectric behaviors of the composite [129]. Koduri Ramam et al. investigated the multiferroic nanocomposites x[BCT-BZT]-(1−x)CoFeCrO, where x ranged from 0.2, 0.4, 0.6, 0.8. As x increased in the multiferroic nanocomposites system, the electrical properties increased up to x = 0.6 and then decreased. In this series, x = 0.6 was found to be optimum with relatively high dielectric constant and low dielectric loss [130]. The microwave assisted radiant heating (MARH) sintered technique to synthesize (BZT–NFO) composite was employed by Arpana Singh et al. They found that the increase in temperature and microwave power percentages increased the dielectric permittivity. About 35% increase in dielectric constant was observed for 15% Mw power sintered sample while *tanδ* increased with increase in temperature and Mw power percentages [131]. The comparison of dielectric properties of different multiferroic composites is described in Table 3. 

### 7.2. Future Prospects

As the dielectric materials possess high power density and large energy density, they are gaining attracted from the scientific community for its use in energy storage devices. Due to large dielectric constant and high breakdown strength, the dielectric polymers and dipolar ceramics are more attractive for energy storage applications. Therefore, studies are focused on enhancing the dielectric constant by structural modification and doping of different materials [132]. As Liu and Ren proposed a critical triple point MPB composition, exploring the morphotropic phase boundary (MPB) system could be an effective method to seek high-performance BCZT materials with excellent dielectric properties to be utilized for energy storage purposes. We also upgraded BCT-BZT to BCT-BMT and achieved excellent dielectric and energy storage properties which appeal the scientific community to try such up gradations [133,134]. Furthermore, these ceramics are sintered at too high temperatures, meaning that their synthesis process is hardly reproducible and the composition homogeneity is rather uncontrollable. Therefore, efforts are required to design such novel synthesis methods which are possible with lower temperatures. The sintering parameters can affect the density, microstructure and dielectric properties of BCZT ceramics. Therefore, more detailed investigation on different sintering conditions and storage energy should be carried out in future studies to obtain a better dielectric response of these materials. Optimizing intrinsic and extrinsic parameters i.e. composition, bulk structure, thickness and grain size can enhance energy densities with improved dielectric breakdown so could be used for commercially available capacitors for energy storage applications, such as actuators etc. [135]. Several theoretical strategies are required to design a material with immense energy storage properties by using BCZT. Due to large power density, good temperature and mechanical stability, fast charging or discharging and good fatigue tolerance, BCZT would be used as dielectric ceramics for energy storage applications in the near future [18,96]. Fabrication of thin films of BCZT by different techniques has been previously reported and still needs further exploration to obtain the dielectric capacitors possessing large energy storage density, high efficiency and high thermal stability for modern electronic devices to operate in harsh environments [62]. The development of single crystals and thin films still lags behind bulk BCZT, so further investigations are needed. The addition of the glass in BCZT exhibited a large dielectric constant, slimmer hysteresis loop, higher polarization and improved remnant polarization. In addition, the energy storage performance of the glass-modified ceramics is far superior to that of the pure ceramic material, so all these findings are attractive enough for researchers to explore the effects of glass addition in BCZT ceramics to obtain improved dielectric properties to be utilized in energy storage applications. Besides the piezoelectric properties or the ferroelectric and ferrimagnetic phase ratios, there are other factors like grain size, density, and dielectric properties which affect the magnetoelectric effect in multiferroic composites. Therefore, these properties of BCZT constituents should be optimized for high-performance lead-free multiferroic magnetoelectric materials.

### 7.3. Conclusions

The BCZT, being an outstanding RFE ceramic, is getting attention because it has a large piezoelectric coefficient. As it is an environmentally friendly material, therefore it has become an auspicious candidate for lead-free energy storage applications. Therefore, the dielectric and energy storage properties were reviewed here to get a better insight for the use of these materials in technology applications. For the use of BCZT, particularly for the energy storage applications by optimizing the dielectric parameters of this material, we concluded that the electric properties are related to phase structure. The MPB of BCZT ceramics are closely related to the presence of an intermediate phase between rhombohedral and tetragonal phases at a narrow region, which could be carefully adjusted by contents of Ca and Zr and by controlling the temperature to obtain enhanced dielectric properties. By varying Zr/Ti ratio, the dielectric constant can be tailored to suitable values for tunable capacitor applications. There is an undeniable link between mean grain size with dielectric properties as the high density of refined grains leads to an improvement of dielectric properties. Introduction of a dopant material has become an important strategy to obtain high performance piezoelectric ceramics by introducing appropriate donors and acceptors in the ceramics. Among different dopings, the addition of *La*_2_*O*_3_ and *Y*_2_*O*_3_ showed a large permittivity (21,736) and lower dielectric loss (0.063) so could be a good candidate material for ceramic capacitor applications. The addition of the glass in BCZT also exhibited a large dielectric constant of 3458 even at 25 °C, slim hysteresis loop with the maximum polarization and reduced grain size with increasing glass concentration. The BCZT thin films lead to a larger dielectric constant and higher dielectric tunability. The charge and discharge energy densities of BCZT thin films are found to be higher than that of bulk counterparts. The change in the substrate leads to different dielectric properties. Experiments have shown that the BCZT thin films are good candidates for lead-free MEMs applications and have also shown inspiring energy storage density up to 99.8 J cm^−3^ and efficiency of 71.0%. Therefore, BCTZ turned out to be an encouraging high dielectric constant material for the tunable devices industry. The BCZT will revolutionize the ceramics industry with its lead-free nature, higher dielectric constant, lower dialectic loss and impressive energy storage properties. 

The authors of the review have been involved in exploring ferroelectric materials for a decade. Both corresponding authors (Prof. Biaolin Peng and Prof. Wenhong Sun) have published more than 60 papers in the ferroelectrics and III-nitrides research fields respectively, and recently co-authored many high reputation publications including those published in Energy and Environmental Science, Journal of Materials Chemistry C, Ceramics International, etc. Some of those have been cited in different sections of this review as reference content. The research work of the group is mainly focused on electrocaloric effect, energy storage, dielectric tunability, piezoelectric, and so on. Therefore, being inspired from our findings, an effort has been performed to review the work of different people of this specific area of ferroelectric materials for energy storage applications and dielectric tunability.

## Figures and Tables

**Figure 1 materials-12-03641-f001:**
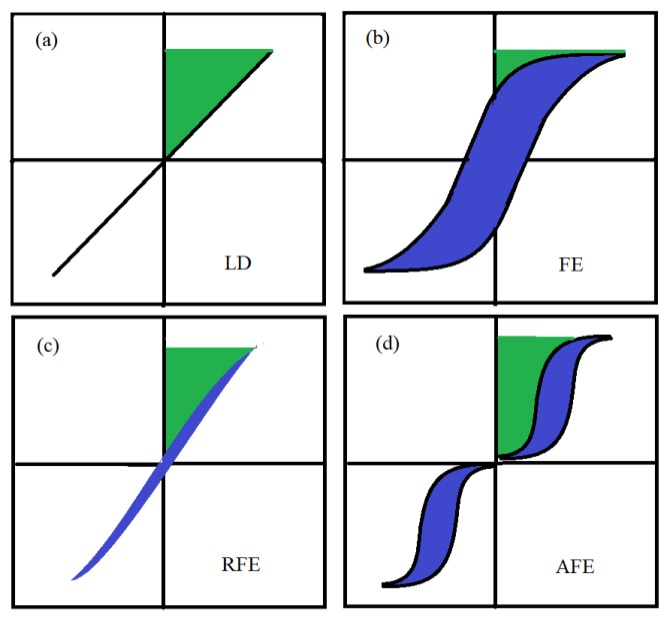
Hysteresis and corresponding energy storage density for (**a**) linear dielectrics, (**b**) ferroelectrics, (**c**) relaxor ferroelectrics, and (**d**) antiferroelectrics. The green area represents energy density and the blue area refers to the energy loss.

**Figure 2 materials-12-03641-f002:**
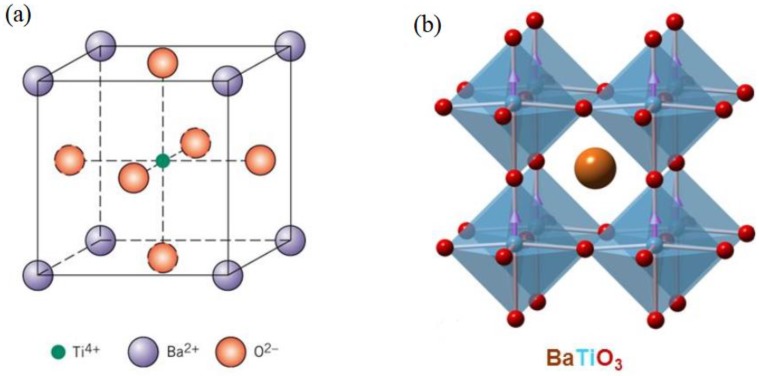
The cubic phase for BaTiO_3_ unit cell (**a**). Here Ba^2+^ ions occupied the corner of the cube, O^2−^ lies in the center of the phases for the composition, Ti^+4^ lies in the center of the composition. 3D view of BaTiO_3_ perovskite crystal structure (**b**), image taken from http://slideplayer.com/slide/9010945/.

**Figure 3 materials-12-03641-f003:**
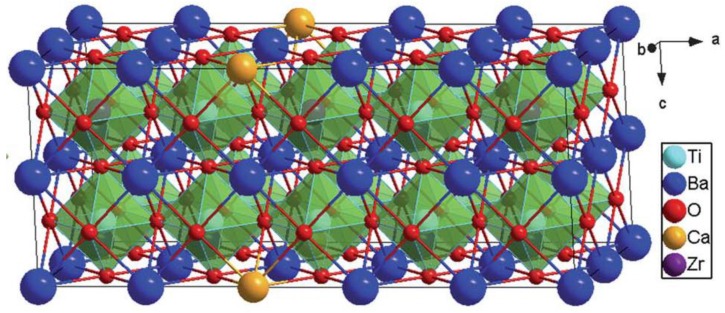
Cubic perovskite structure and arrangements of atoms in a B_0.95_C_0.05_Z_0.15_T_0.85_O_3_ supercell with 5 × 2 × 2 dimension. Reprinted with permission from [54]. Copyright 2012 Royal Society of Chemistry.

**Figure 4 materials-12-03641-f004:**
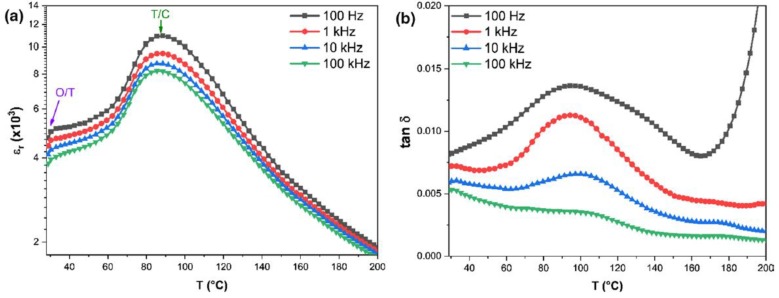
Temperature-dependence of the (**a**) dielectric constant and (**b**) dielectric loss of Ba_(1−x)_Ca_x_Zr_y_Ti_(1−y)_O_3_ (BCZT) ceramic at different frequencies. O/T corresponds to orthorhombic–tetragonal transition and T/C denotes the tetragonal–cubic phase transition. Reprinted with permission from [55]. Copyright 2019 Springer Nature.

**Figure 5 materials-12-03641-f005:**
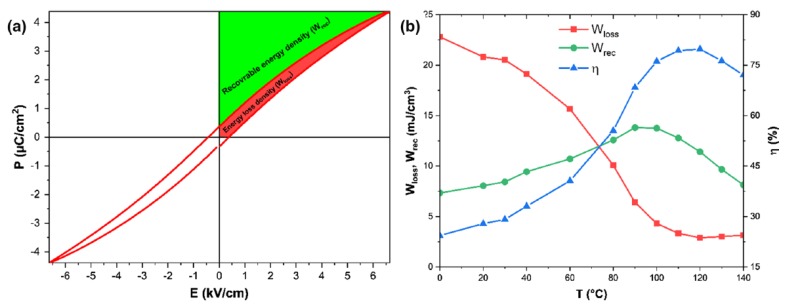
P–E loop at 120 °C with schematic calculations energy storage efficiency (**a**) and variation of recoverable energy density, energy loss density, and the energy storage efficiency of the BCZT ceramic with temperature (**b**). Reprinted with permission from [55]. Copyright 2019 Springer Nature.

**Figure 6 materials-12-03641-f006:**
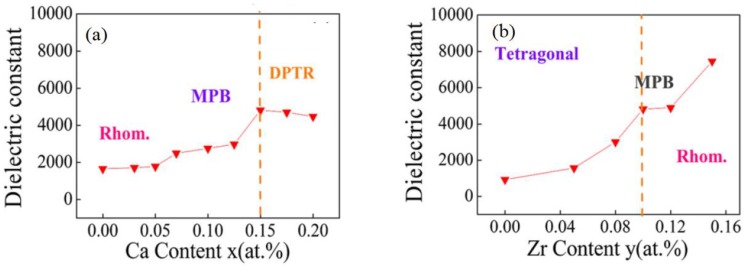
(**a**) The dielectric constant (*ε_r_*) of (Ba_1_x_Ca_x_)(Zr_0.1_Ti_0.9_)O_3_ ceramics. (**b**) The dielectric constant (*ε_r_*) of (Ba_0.85_Ca_0.15_)(Zr_y_Ti_1_y_)O_3_ ceramics. For different acronyms in the figures, please see the text. Reprinted with permission from [57]. Copyright 2012 The American Ceramic Society.

**Figure 7 materials-12-03641-f007:**
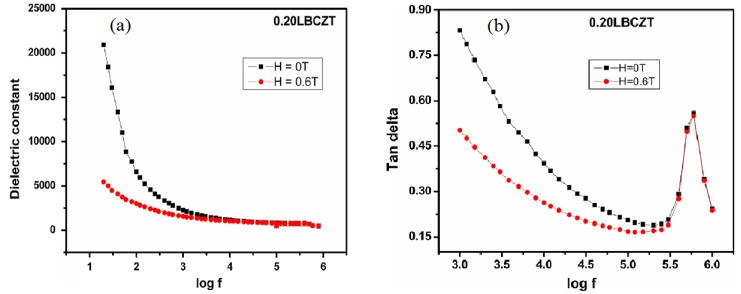
(**a**) Variation of ε as a function of logf for 0.20LBCZT Heterostructure. (**b**) Variation of *tanδ* as a function of logf for 0.20LBCZT with and without applied magnetic field (H). Reprinted with permission from [51]. Copyright 2016 Springer Nature.

**Figure 8 materials-12-03641-f008:**
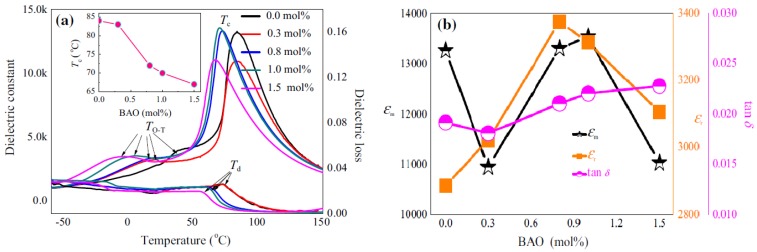
The temperature dependence of *ε_r_* and *T_C_* for (1−x)BCZT–xBAO ceramics measured at 1 kHz (**a**). The maximum dielectric constant (ε_m_), dielectric constant (*ε_r_*), and dielectric loss (*tanδ*) of BCZT ceramics with different BAO contents (**b**). Reprinted with permission from [76]. Copyright 2015 Springer Nature.

**Figure 9 materials-12-03641-f009:**
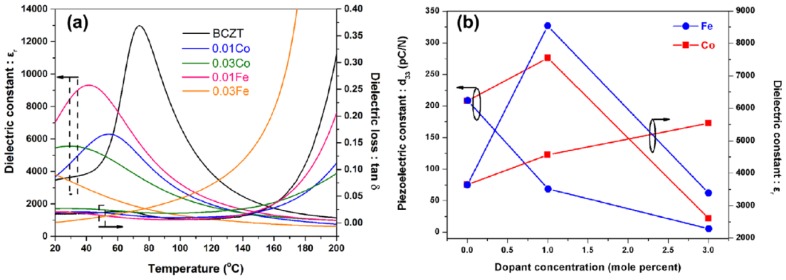
Dielectric and piezoelectric properties of BCZT-xCo and BCZT-xFe ceramics. Dielectric versus temperature at 1 kHz (**a**) and piezoelectric constant as a function of dopant concentration (**b**). Reprinted with permission from [71]. Copyright 2018 WILEY-VCH Verlag GmbH & Co.

**Figure 10 materials-12-03641-f010:**
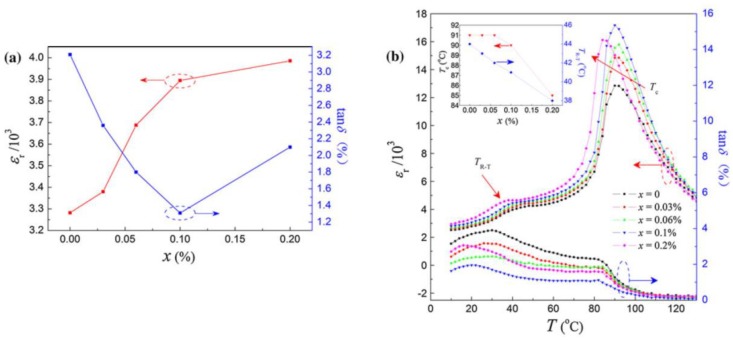
(**a**) Dielectric properties as a function of x at 1 k Hz and (**b**) temperature dependence of the *ε_r_* and *tanδ* of BCZT-xSb ceramics. Reprinted with permission from [78]. Copyright 2011 Springer Nature.

**Figure 11 materials-12-03641-f011:**
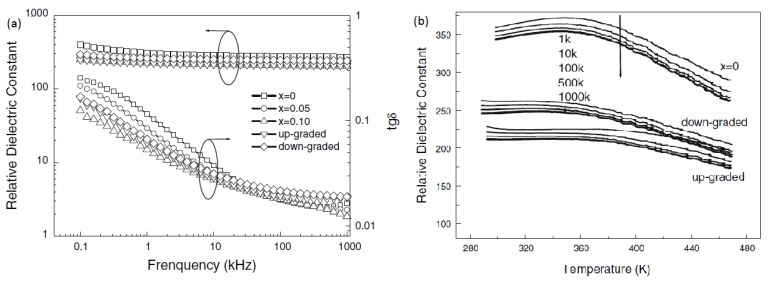
The dielectric constant and dielectric loss of the homogenous and compositionally graded BCZT thin films as functions of frequency, measured at room temperature (**a**). The dielectric constant of Ba_1−x_Ca_x_Zr_0.05_Ti_0.95_O_3_ (x = 0) and compositionally graded BCZT thin films as a function of temperature (**b**). Reprinted with permission from [87]. Copyright 2008 Springer Nature.

**Figure 12 materials-12-03641-f012:**
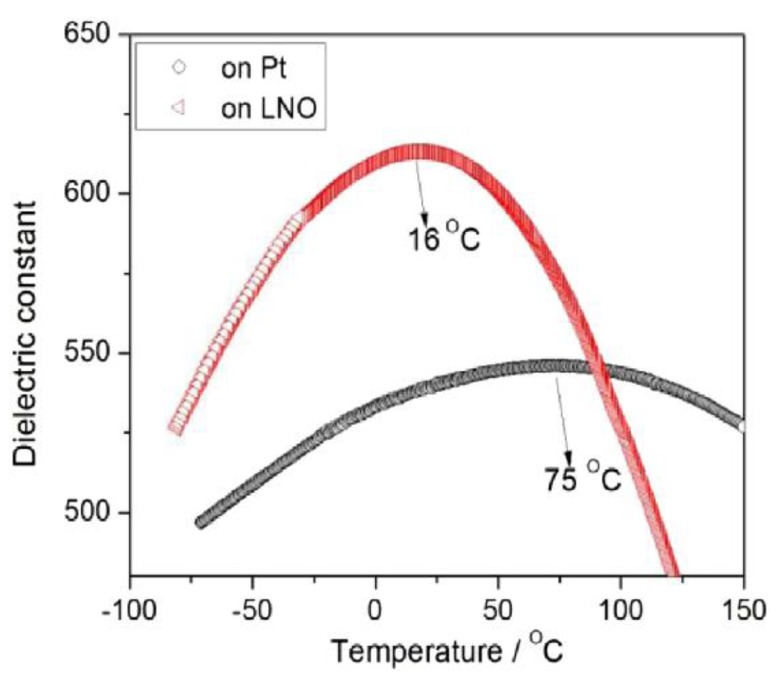
Dielectric constant as a function of temperature measured at frequency of 100 kHz for the BCZT thin films deposited on Pt(111)/Ti/SiO2/Si substrate and LNO/Pt(111)/Ti/SiO_2_/Si substrate. Reprinted with permission from [60]. Copyright 2013 Elsevier.

**Figure 13 materials-12-03641-f013:**
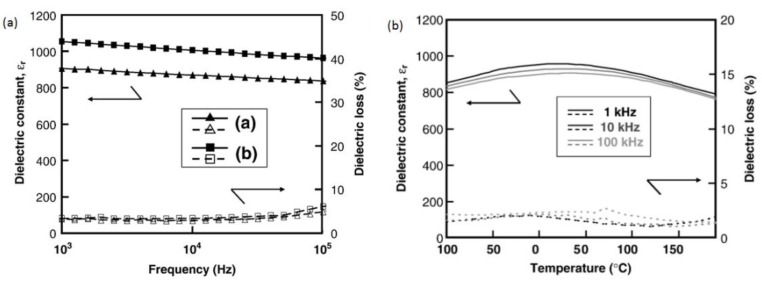
Dielectric constant and loss of (Ba,Ca)(Zr,Ti)O_3_ thin films on Pt/TiOx/ SiO_2_/Si substrates crystallized at 700 °C as a function of the measured frequency, (**a**) Ba:Ca = 1.0:0.0 (▲, △) and (**b**) Ba:Ca = 0.95:0.05 (■, □) [Zr:Ti = 0.20:0.80] (**a**). Temperature dependence of dielectric constant and loss of (Ba,Ca)(Zr,Ti)O_3_ thin film on Pt/TiOx/SiO_2_/Si substrate crystallized at 700 °C (measured frequency: 1–100 kHz) [Ba:Ca = 0.95:0.05, Zr:Ti = 0.20:0.80] (**b**). Reprinted with permission from [90]. Copyright 2008 Elsevier.

**Figure 14 materials-12-03641-f014:**
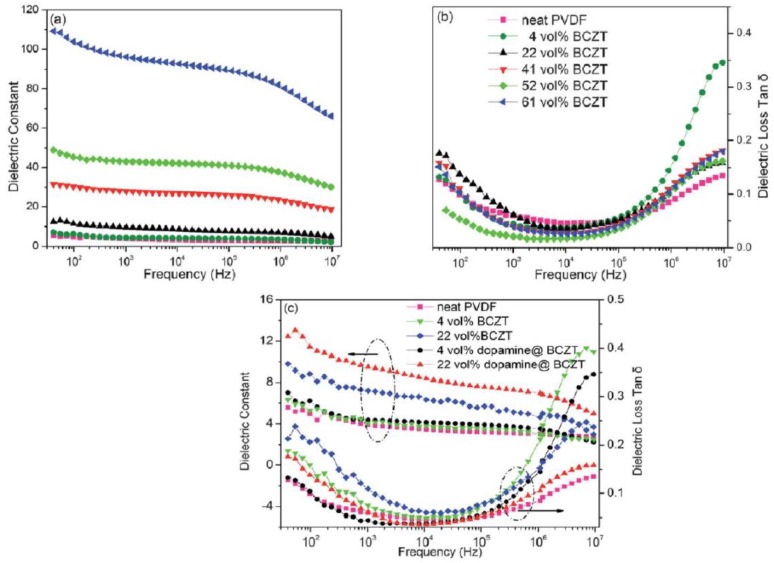
Frequency dependence of the dielectric constant (**a**) and the dielectric loss tangent, (**b**) of dopamine@BCZT/PVDF composite films, and (**c**) BCZT/PVDF (Poly(vinylidiene fluoride) composite films. Reprinted with permission from [54]. Copyright 2012 Royal Society of Chemistry.

**Figure 15 materials-12-03641-f015:**
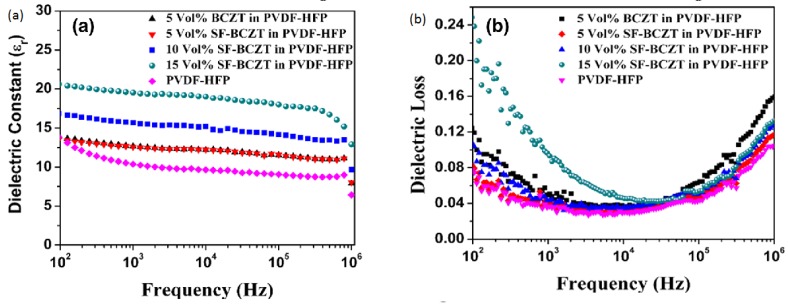
Dielectric constant versus frequency (**a**), dielectric loss versus frequency (**b**) for different molar concentration of PVDF-HFP in BCZT. Reprinted with permission from [61]. Copyright 2018 Springer Nature.

**Figure 16 materials-12-03641-f016:**
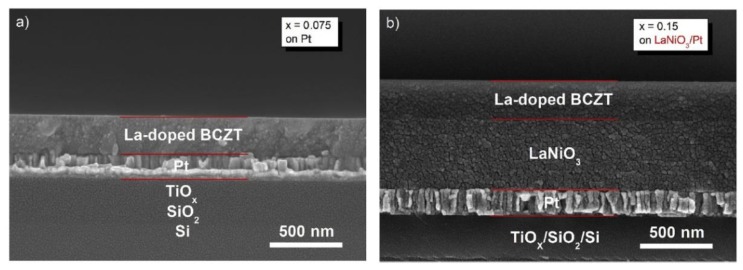
Cross-sectional SEM images of La-doped BCZT thin films at x = 0.0075 and 0.15. (**a**) On Pt bottom electrodes, and (**b**) on LaNiO_3_/Pt composite bottom electrodes. Reprinted with permission from [62]. Copyright 2019 Elsevier.

**Figure 17 materials-12-03641-f017:**
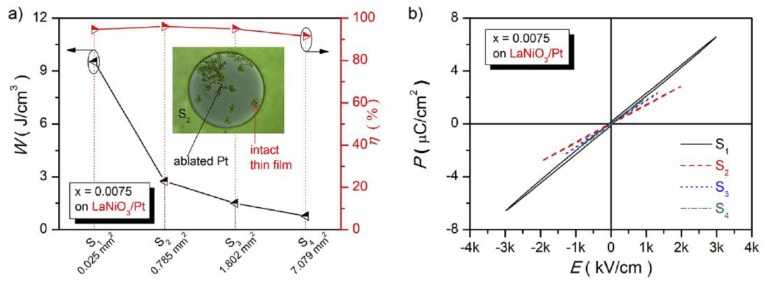
La-doped BCZT thin films at x = 0.0075 on LaNiO_3_/Pt composite bottom electrodes. The area of top electrode as a function of W and η. Inset: image of Pt top electrode (**a**). Polarization-electric field (P-E) loops with various sizes of Pt top electrodes (**b**). Reprinted with permission from [62]. Copyright 2019 Elsevier.

**Figure 18 materials-12-03641-f018:**
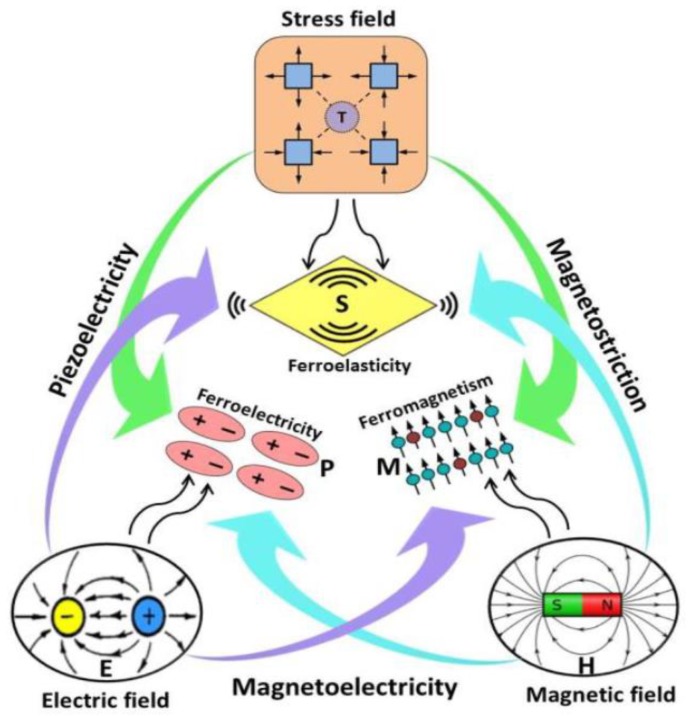
Schematic illustration of magnetic-elastic-electric couplings in multiferroic materials. Here, M is magnetization, S is mechanical strain, and P is dielectric polarization. Reprinted with permission from [103]. Copyright 2019 MDPI Publishers.

**Figure 19 materials-12-03641-f019:**
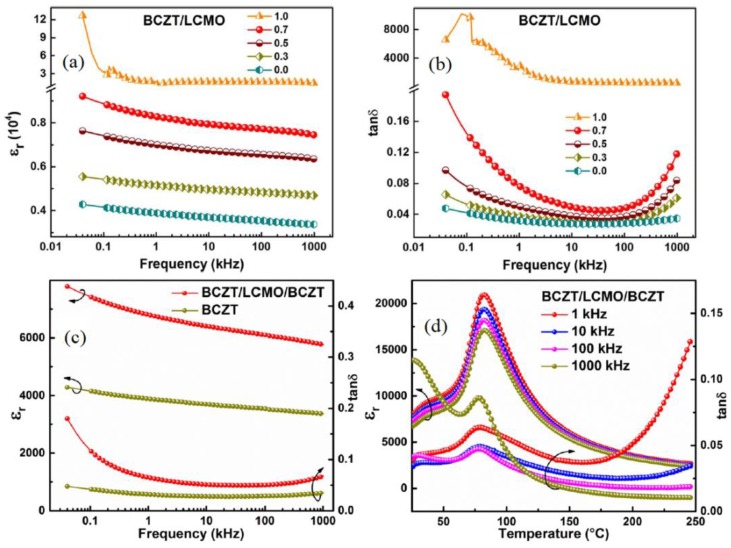
Room temperature frequency dependence of *ε_r_* (**a**) and *tanδ* (**b**) for the (1−x)BCZT-xLCMO laminated composites and frequency (**c**) and temperature dependence (**d**) of *ε_r_* and *tanδ* for BCZT-LCMO-BCZT composite. Reprinted with permission from [121,122]. Copyright 2017 Elsevier and 2019 AIP Publishing.

**Table 1 materials-12-03641-t001:** The comparison of dielectric constant and dielectric loss for different dopants.

Dopant	Content (mol%)	*ε_r_*	*tanδ*	Ref
BAO	0.8	3375	0.021	[76]
CuO	0.02–0.10	3450–3400	1.25–1.375	[66]
La_2_O_3_/Y_2_O_3_	0–0.5	16,817–21,736	0.063–0.1062	[29]
Co	0.01–0.03	4567–5540	--	[71]
Fe	0.01–0.03	8543–3394	---	[71]
P(VDF-TrFE)	0–0.4	11.2–90.7	0.024	[77]
Sb_2_O_3_	0.2	4958	1.8	[78]
Sb_2_O_3_	0.1	3895	1.3	[79]
Nb^5+^	0.0–1.0	2103–4636	0.0130–0.0326	[80]
V_2_O_5_	0–0.50	2175–3104	0.030–0.043	[50]
Nd^3^^+^	0–0.02	17,184–6095	0.0245–0.0119	[81]
Mn	0–2.0	1314–1651	2.40–2.99	[52]

**Table 2 materials-12-03641-t002:** Comparison of dielectric and energy storage properties of BCZT thin films.

Substrate	Process	*ε_r_* (pm/V)	*tanδ*	Frequency	Temperature	Pr	Energy Storage	DBS	Ref
Pt/Ti/SiO_2_/Si	Sol–gel	50	(0.02)	100 kHz	75 °C	−	−	−	[60]
Si	Sputter-deposited	−	Higher dielectric loss	20 Hz-1 MHz	700–900 °C	−	~8 × 10^−3^ W-h/kg	200 kV/cm ^−2^ MV/cm	[88]
Al	Sol–gel	9.6	Lower dielectric loss	10 KHz	200 °C	−	8.5 J cm^−3^	(>300 kV mm^−1^)	[61]
Pt(111)/Ti/SiO_2_/Si	Solid state reaction	601–2347	Increased dielectric loss	100 to 10 KHz	330 °C	47–11.8 µC cm^2^	99.8 J cm^−3^	−	[89]
Pt/TiOx/SiO_2_/Si	Thin-film processing methods	900	5%	−	700 °C	−	−	−	[90]
Pt/Ti/SiO_2_/Si substrates without CaRuO3 (CRO) buffer layer	Pulsed laser deposition (PLD)	725	Decrease from 0.036	100 Hz to 1 MHz	−	−	2 J/cm^2^	−	[91]
Pt/Ti/SiO_2_/Si substrate with CaRuO3	Pulsed laser deposition	877	0.023	100 Hz to 1 MHz	−	−	2 J/cm^2^	−	[91]
Pt/Ti/SiO_2_/Si	Sol–gel under three different annealing processes	Decreases slowly for 8000–60,000 Hz	Dielectric loss is smaller for higher frequencies	10,000–60,000 Hz	700 °C	10.08 μC/cm^2^	−	−	[93]
Pt/Ti/SiO_2_/Si	Sol–gel	550	(0.02)	100 KHz	75 °C	−	−	−	[60]
LaNiO_3_(LNO)/Pt/Ti/SiO_2_/Si substrates	Sol–gel	620	(0.04)	100 KHz	16 °C	−	−	−	[60]
Clean glass plates	Solution casting method	~140	1.8 to 0.2	Frequency range of 60–10^7^ Hz	474–497 °C	0.206 μC/cm^2^	5.3 J/cm^3^	20–96 kV mm^−1^	[98]
Flexible polymeric substrates STO (001)	PLD deposition process description	2600–2700	~0.02	1 kHz to 1 MHz	−	−	−	−	[94]
(100)-SrRuO_3_/SrTiO_3_ substrate	Pulsed laser deposition	400	~0.3	f = 1.6 kHz	At room temperature	35.79 and 12.43 μC cm^−2^	−	−	[99]
Pt/MgO	Metal organic decomposition (MOD)	504	<0.04	1 MHz	600–900 °C	−	−	−	[95]
LaNiO_3_/Pt	Sol–gel method	−	_	−	0–300 °C	−	15.5 J/cm^3^	(>1000 kV/cm)	[62]

**Table 3 materials-12-03641-t003:** Comparison of dielectric properties of different multiferroic composites.

Ferroelectric Constituent	Ferromagnetic Constituent	Remnant Polarization	Dielectric Constant	Dielectric Loss	Ref.
BZT	LSM	15.7	10,516	Increase with LSM content	[130]
BCT-BZT	CNFO	-	2264	2.58	[129]
NBT	NMF	NBT	714	0.03	[118]
BCZT	LCMO	9.4	20,280	0.04	[122]
BTO	CFO	4.46	-	-	[112]
NBT	ZFO		Decreases at lower frequencies	Decreases at lower frequencies	[117]
BTO	BTF doped with Co and Ni	$2P_{r}=7.23$	Enhanced by the application of magnetic field	Enhanced by the application of magnetic field	[115]
BZT-BCT	LSMO	24	~16,000	-	[125]
BCZT	CFO	-	~1800	-	[127]
BCZT	NFO	8.34	-	-	[121]
BCZT	CFO	~1.4	~678	0.093	[120]
BZT-BCT	LSMO	~47	4900–5100	0.02	[126]
BZT	NFO	-	~11,000	Increases with increasing temperature and Mw power percentages	[132]
BZT-BCT	CFO	-	2489	0.0542	[131]
BTO	CFO	-	~3.3 × 10^4^	~0.19	[111]
BNT-BT	NCZF	-	Decreases with increasing frequency	Decreases with increasing frequency	[113]
BT	NZF	30	2250	0.3	[116]
BCZT	LCMO	7.444	~$10^{4}$	Decreases at low frequency and increases at high frequency	[123]
BST	NCF	-	Dielectric constant decreases	Decreases with increasing frequency	[114]
BCT-BZT	LSMO	1.57	~2500	~0.02	[128]
PZT	CFO		~540	~0.01 at 1 MHz	[110]

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
