# Peer review of "Dielectric and Energy Storage Properties of Ba(1−x)CaxZryTi(1−y)O3 (BCZT): A Review"

_materials, 2019, doi:10.3390/ma12213641_

Round 1
Reviewer 1 Report
This review report present a detailed summary and outlook of BZT-BCT based dielectric ceramics for high energy storage capacitors and other ferroic-based applications.
Following comments are needed to be address before accepting the paper for publishing ;
Author by-line is incomplete! Define the acronym at first place, like BAO on page#8 etc. Figure captions are very short and incomplete. Example, Fig.6 what is DPTR in (a)? Figure caption should have all the details about authors want to convey the readers. If you are allowing readers to check and read the cross-references then what is the point to write a review article! English and typos are needed to be correct through-out the manuscript. Example, authors use two spellings "polarisation" and "polarization" at page#2. Add the following relevant reference in section 3.3 "modified BTO", which deals with the doping and solid-solution induced disorder in BTO-based materials- https://doi.org/10.1103/PhysRevMaterials.2.060404Author Response
Thank you very much for your comments. We have revised the manuscript as follows:
Comment 1: Author by-line is incomplete:
Response: Very sorry for this major line mistake. It has been fixed in the revised manuscript.
Comment 2: Define the acronym at first place, like BAO on page#8 etc
Response: The manuscript is being read carefully again and all the acronyms have been explained in the associated part of the manuscript.
Comment 3: Figure caption should have all the details about authors want to convey the readers.
Response: Figure captions are being read carefully and necessary information which is needed to convey the message properly has been added.
Comment 4: English and typos are needed to be correct through-out the manuscript. Example, authors use two spellings "polarisation" and "polarization" at page#2.
Response: The manuscript is being read carefully again and all the typos and grammar issues have been corrected in the revised manuscript.
Comment 5: Add the following relevant reference in section 3.3 "modified BTO", which deals with the doping and solid-solution induced disorder in BTO-based materials- https://doi.org/ 10.1103/ PhysRevMaterials.2.060404
Response: The said paper has been added as per suggestion in the defined section of the revised manuscript.

Reviewer 2 Report
The manuscript titled "Dielectric and energy storage properties of Ba(1-x)CaxZryTi(1-y)O3(BCZT): A Review" by Wenhong Sun et al., is a comprehensive review on the current status and future prospects of the ceramic BCZT materials for Lead-free energy storage applications. The manuscript is well organized and referred to the most appropriate and adequate references.
It would be interesting to add a paragraph in the manuscript on the authors´research experience in this area and their contribution that draws the motivation to write a review on the topic?
Carefully check and correct the list of authors line in the manuscript which is ending with “and”.
Carefully check and correct the language and grammatical errors throughout the manuscript.
Author Response
Thank you very much for your comments. We have revised the manuscript (attached) as follows:
Comment 1: It would be interesting to add a paragraph in the manuscript on the authors research experience in this area and their contribution that draws the motivation to write a review on the topic?
Response: The authors of this review are involved in the research work of ferroelectric materials since past few years and have published high impact peer reviewed papers. So motivated by their outcomes and fascinated by this emerging field we plan to write this review to provide the scientific community a draft for consideration before designing their project under the light of information provided in this review. So a paragraph about the author’s scientific contribution to this field is added in the revised manuscript as follows:
The authors of the review are involved in exploring the ferroelectric materials since a decade. Both corresponding authors (Prof. Biaolin Peng and Prof. Wenhong Sun) have published more than 60 papers in the ferroelectrics and III-nitrides research fields, respectively, and recently co-authored many high reputation publications including the Energy & Environmental Science, Journal of Materials Chemistry C, Ceramics International, etc.. Some of them have been cited at different sections of this review as reference content. The research work of the group is mainly focused on electrocaloric effect, energy storage, dielectric tunability, piezoelectric and so on. So being inspired from our findings, an effort has been performed to review the work of different people of this specific area of ferroelectric materials for energy storage applications and dielectric tunability..
Comment 2: Carefully check and correct the list of authors line in the manuscript which is ending with “and”.
Response: Very sorry for this major line mistake. It has been fixed in the revised manuscript.
Comment 3: Carefully check and correct the language and grammatical errors throughout the manuscript
Response: The manuscript is being read carefully again and all the typos and grammar issues have been corrected in the revised manuscript.

Reviewer 3 Report
This is a good review on an important lead-free ferroelectric system. I recommend to publish.
The manuscript is a review article, and the authors have gathered and summarized info based on the published literature. I find the article is fairly organized to be published.
I am afraid that I cannot be more helpful on this occasion.
Author Response
Thank you very much for reviewing our paper.